# Biotic and abiotic factors affecting Atlantic ghost crab (*Ocypode quadrata*) spatiotemporal activity at an important shorebird nesting site in Virginia

**Mikayla N. Call**[1][☉]*, **Rasheed S. Pongnon**[1][☉], **Christy N. Wails**[1][‡], **Sarah M. Karpanty**[1][‡], **Kristy C. Lapenta**[1], **Alexandra L. Wilke**[2], **Ruth Boettcher**[3], **Camille R. Alvino**[1], **James D. Fraser**[1]

1 Department of Fish and Wildlife Conservation, Virginia Tech, Blacksburg, Virginia, United States of America, 2 The Nature Conservancy in Virginia, Nassawadox, Virginia, United States of America, 3 Virginia Department of Wildlife Resources, Machipongo, Virginia, United States of America

☉ These authors contributed equally to this work.
‡ CNW and SMK also contributed equally to this work.
* mncall@vt.edu

**Data Availability Statement:** All data are available from the Environmental Data Initiative Data Portal (https://doi.org/10.6073/pasta/

## Abstract

Atlantic ghost crabs (*Ocypode quadrata*) are predators of beach-nesting shorebird nests and chicks on the United States' Atlantic and Gulf coasts. Ghost crabs may also disturb birds, altering foraging, habitat use, or nest and brood attendance patterns. Shorebird conservation strategies often involve predator and disturbance management to improve reproductive success, but efforts rarely target ghost crabs. Despite the threat to shorebird reproductive success, ghost crabs are a poorly understood part of the beach ecosystem and additional knowledge about ghost crab habitat selection is needed to inform shorebird conservation. We monitored ghost crab activity, defined as burrow abundance, throughout the shorebird breeding season on Metompkin Island, Virginia, an important breeding site for piping plovers (*Charadrius melodus*) and American oystercatchers (*Haematopus palliatus*). We counted burrows at shorebird nests and random locations throughout the breeding season and investigated whether ghost crab activity was greater at nest sites relative to random locations without shorebird nests. While we observed burrows at all nest sites ($n = 63$ nests), we found that burrow counts were lower at piping plover nests with shell cover, relative to random locations with no shell cover. Ghost crabs may avoid piping plover nest sites due to anti-predator behaviors from incubating adults or differences in microhabitat characteristics selected by piping plovers. We also investigated the effects of habitat type, date, and air temperature on the abundance of ghost crab burrows. We found that while crab burrows were present across the barrier island landscape, there were more burrows in sandy, undisturbed habitats behind the dunes, relative to wave-disturbed beach. Additionally, ghost crab activity increased later in the shorebird breeding season. Understanding when and where ghost crabs are most likely to be active in the landscape can aid decision-making to benefit imperiled shorebird populations.

1f1117a917de507802f1acb4d5cf42ef; Call et al. 2024).

**Funding:** This research was supported by the National Science Foundation (NSF) Virginia Coast Reserve Long Term Ecological Research Grant DEB-1832221 (https://www.nsf.gov/; SMK, RSP), the Virginia Sea Grant Graduate Research Fellowships (https://vaseagrant.org/; MNC), the Virginia Tech College of Natural Resources and Environment (https://cnre.vt.edu/; RSP), and the Virginia Tech Department of Fish and Wildlife Conservation (https://fishwild.vt.edu/; RSP). Any opinion, findings, and conclusions or recommendations expressed in this material are those of the authors and do not necessarily reflect the views of the National Science Foundation, or of other sponsors. The funders had no role in study design, data collection and analysis, decision to publish, or preparation of the manuscript.

**Competing interests:** The authors have declared that no competing interests exist.

## Introduction

Ghost crabs *(Ocypode* spp.) fill a broad trophic niche, preying on and scavenging items from small organic particles and plant detritus to larger beach-dwelling vertebrates [1]. On the United States' Atlantic and Gulf coasts, Atlantic ghost crabs (*O. quadrata*; hereafter, 'ghost crab,') prey on the eggs and young of imperiled beach-nesting species [1], including sea turtles (Families Cheloniidae and Dermochelyidae) [2] and shorebirds (Order Charadriiformes) [3–7]. Additionally, ghost crab presence may indirectly affect the survival of the nests and young of imperiled species by creating a disturbance that can attract other predators [8] or, in the case of shorebirds, cause the adults to leave the nest or chick-rearing site [9]. Ghost crabs may also cause injury to adult shorebirds [5], which could have an indirect effect on the survival of the adults, their nest, or chicks.

As shorebird populations rapidly decline [10], beach managers have attempted to address factors limiting population growth, including the threat of predation to shorebird reproductive success [11]. On the Western Atlantic Flyway, predation of nests and pre-fledged chicks is managed through removal of mammalian predators (e.g., red fox *Vulpes vulpes*, northern raccoon *Procyon lotor*, and coyote *Canis latrans*) and use of predator exclusion devices that restrict larger mammalian and avian predators from accessing a nest [12–14]. However, despite these efforts, predation threats to nest and chicks may remain when ghost crabs are present [4]. Ghost crabs are not often targeted for removal and are typically capable of entering predator exclusion devices [4, 13]. While there is some evidence that the lethal removal of ghost crabs can improve shorebird reproductive success within the Western Atlantic Flyway (e.g., [15]), ghost crab management generally does not surface as a leading conservation strategy and warrants more attention from the conservation community.

The threat of ghost crab activity to breeding shorebirds is evolving as ghost crab populations respond to rapid, climate-driven change within coastal ecosystems. For example, warming ocean temperatures are allowing *Ocypode* spp. to expand northward from their former range limits [16]. In the northeastern United States, where a warming hotspot has caused increased sea surface temperatures [17], ghost crabs appear to be expanding into previously unoccupied northern latitudes [18], presenting a new threat to nesting shorebirds. Additionally, ghost crab population abundance may change rapidly in response to anthropogenic activities such as development, beach renourishment, or predation management [19]. Ghost crab populations may increase where their mammalian predators are removed to benefit imperiled breeding shorebirds, as has been observed for breeding sea turtles [8]. Thus, ghost crabs may be an under-realized threat to shorebird conservation and broad scale ghost crab population management may be needed.

Understanding drivers of ghost crab activity at shorebird breeding sites will help predict when and where ghost crabs are likely to be most active, thus informing management and providing insight into how the relative threat of ghost crabs to breeding shorebirds may be impacted by climate change. We identified factors affecting the spatial and temporal patterns of ghost crab activity at an important shorebird breeding site within the Virginia barrier island system. We also investigated whether ghost crab activity appeared to be greater at shorebird nest sites relative to other locations within the study area. Given the role of ghost crabs as a predator of shorebird nests and chicks, we predicted that ghost crabs would select habitat closer to shorebird nest sites than random locations.

## Methods

### Study site

Metompkin Island (37.75˚ N, 75.54˚ W; Fig 1) is an undeveloped barrier island located within the northern extent of the Virginia barrier island system. The island is oriented parallel to the shoreline of the Delmarva Peninsula and is undergoing rapid landward retreat [20, 21]. Metompkin Island is approximately 10 km long, and its sandy beach is frequently disturbed by high-water events [22, 23]. Shell armoring from oyster and clam shells along the beach is common [22]. The beach is backed by salt marsh and tidal flats in the northern part of the island and a discontinuous (i.e., punctuated by overwash areas) dune system that protects a grass and shrub community on the southern part of the island.

The sandy beach and overwash areas on Metompkin Island provide critical breeding habitat for several shorebird species, including piping plovers (*Charadrius melodus*) and American oystercatchers (*Haematopus palliatus*). Both piping plovers and American oystercatchers are species of conservation concern, with the Atlantic coast population of the piping plover federally listed as threatened in the United States [11, 26]. Within the Commonwealth of Virginia, piping plovers and American oystercatchers are Species of Greatest Conservation Need and piping plovers are state listed as threatened [27]. Ghost crabs have been observed using the dune, beach, and intertidal habitats on Metompkin Island throughout their annual life cycle, and camera evidence from Metompkin Island suggests that they are a predator of shorebird eggs [28].

The Nature Conservancy in Virginia and the U.S. Fish and Wildlife Service manage Metompkin Island as an important shorebird breeding and migratory stopover site. Management actions include removal of mammalian predators, restrictions on public access above the high-tide line, limiting public recreational use to diurnal low-impact activities, and weekly monitoring of shorebird reproductive success and threats by trained biologists. These managing entities do not currently monitor or manage the ghost crab population on the island.

We conducted all research activities within a 3-km study area located at approximately the center of Metompkin Island (Fig 1; northernmost extent approx. 37.75˚ N, 75.55˚ W; southernmost extent approx. 37.73˚ N, 75.57˚ W; centroid approx. 37.74˚ N, 75.56˚ W). This study area allowed us to sample a variety of habitat types on Metompkin Island, including beaches backed by salt marsh and tidal flats, and beaches backed by a dune system and stable vegetation community. Both piping plovers and American oystercatchers nested within the study area.

### Field methods

We surveyed ghost crab activity within the study area on Metompkin Island by repeatedly conducting crab burrow counts. We used burrow counts, as opposed to trapping and physical collection of individuals [29], to limit the level of disturbance to breeding shorebirds and their habitat during surveys. We did not assume ghost crab burrow counts to be a direct measure or index of population abundance or density due to the potential for burrow sharing, burrow abandonment, and individuals constructing multiple burrows or burrow entrances [29–31]. Instead, we consider burrow counts to be a measure of ghost crab burrowing activity at a sampling location.

We conducted surveys from April through July 2022. Survey sampling locations included all piping plover (Fig 2A) and American oystercatcher (Fig 2B) nests within the study area and an additional 38 random sampling locations (Fig 2C) located between the high tide line and marsh or stable interior vegetation. The number of random sampling locations that we used was based on the number of nest attempts by American oystercatchers within the study area the previous year (i.e., 2021). We conducted nest surveys at each nest approximately every 1–3

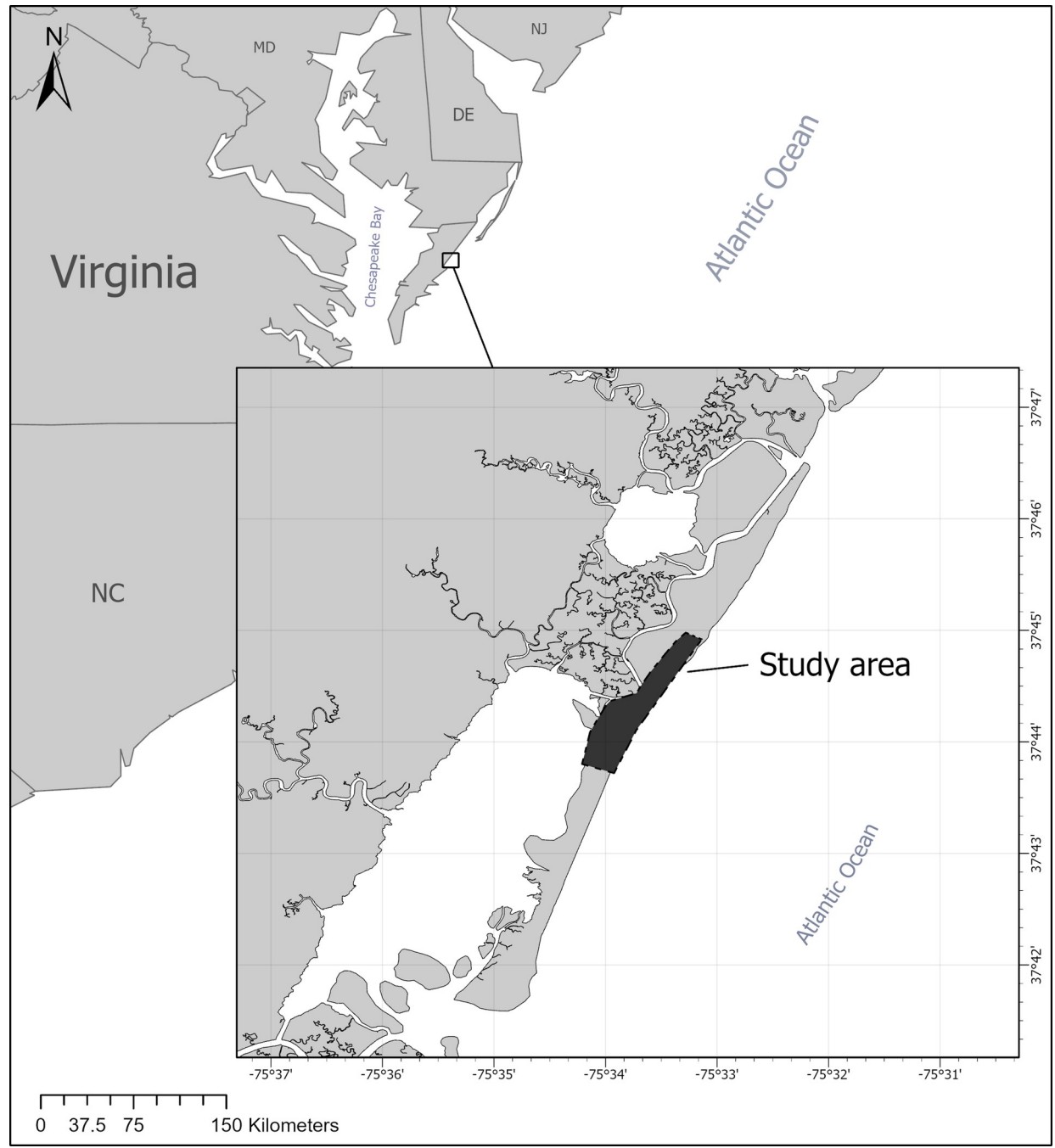

**Fig 1. Location of Metompkin Island, Virginia, and the study area on Metompkin Island.** Basemaps provided by the U.S. Census Bureau [24] and Virginia Geographic Information Network [25].

days, though survey intervals varied due to weather constraints, and at random locations surveys once per week.

We counted crab burrows within a 2 m radius of each nest or random sampling point (Fig 2D) during each visit. The center of the sampling point for each nest was located in the center of the clutch. Both piping plover and American oystercatcher nests consist of clutches of eggs laid in a shallow scrape in the sand, and thus the area surveyed around each nest was the same

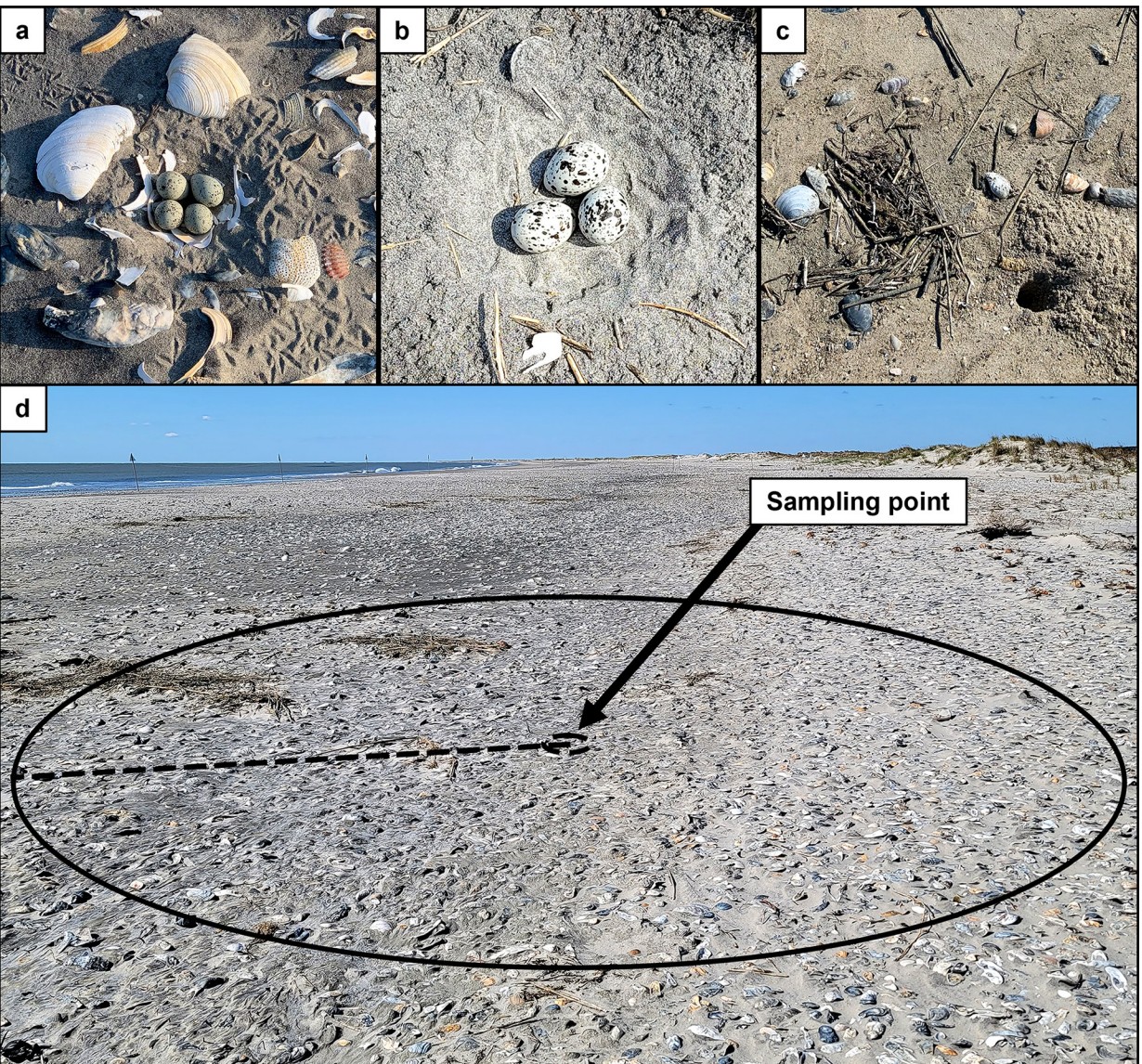

**Fig 2. Ghost crab burrow surveys were conducted at piping plover nests, American oystercatcher nests, and random sampling locations on Metompkin Island, Virginia.** The survey locations in this study included piping plover nests (a), American oystercatchers nests (b), and random locations on the beach that did not include a shorebird nest (c). We counted all ghost crab burrows located within a 2 m radius of each sampling location (d; figure is a graphical representation of the field methods and approximates scale).

as the area surveyed around each random sampling location. Beach-nesting shorebird nests are not large in size, approximately 10 cm in diameter for piping plovers [32] and 20 cm in diameter for American oystercatchers [33]. Additionally, nests did not affect our ability to detect ghost crab burrows, as we were able to count burrows even if they were located within the shallow nest scrape. As we were interested in the activity of ghost crabs capable of predating shorebird eggs or chicks, we only counted crab burrows that were 2.5 cm or larger. To limit potential disturbance to shorebirds, we often used only a visual estimate to determine burrow size. However, if a burrow was not obviously above or below the 2.5 cm threshold, we would measure the burrow entrance to determine if the burrow would be included in the count. We counted all crab

burrows above the size threshold as long as the entrance was not collapsed, as burrows that lack signs of activity (e.g., tracks, excavated sand) may still be occupied [34].

We also recorded a visual estimate of shell density within the area of the 2 m radius (defined as 'none,' 0%; 'sparse,' 1–49%; and 'heavy,' ≥ 50%) and the habitat type (i.e., intertidal zone, berm, beach, dunes, backdune flat, and backbarrier) at each sampling point. We defined the intertidal zone as the portion of the oceanward shore that is inundated regularly at high tide; the berm as a sand ridge positioned above the high-tide line and parallel to the shoreline that forms as a result of sediment deposition by wave runup; the beach as the open, sparsely vegetated sandy habitat between the berm and dunes; the dunes as ridges of accumulated sand that form along the backside of the beach; the overwash as a break in the dune ridge where sediment from the beach has been deposited across and onto the island backside, resulting in open and sparsely vegetated sand habitat; the backdune flat as the sparsely vegetated habitat located behind the dune ridge; and the backbarrier as the stable vegetation or marsh community located on the bayside of the barrier island. We repeated shell density measurements and habitat type classifications once every three weeks at all random sampling locations throughout the breeding season to account for changes precipitated by storms, tides and other factors contributing to the dynamic nature of Metompkin Island. These data were only recorded once for each nest, as incubation typically only lasts between 3–4 weeks for American oystercatchers and piping plovers [32, 33]. All data are available from [35].

## Statistical analyses

We modeled ghost crab burrow abundance as a function of a set of variables that we hypothesized to be important predictors of ghost crab activity, based on prior knowledge of ghost crab ecology [1]. These variables included: date, to account for variation throughout the breeding season; mean daily air temperature (collected from the National Weather Service's station for the Wallops Island area, located approximately 23 km from the study site at 37.94˚ N, -75.47˚ W; available at [36]), to assess the effect of temperature as ghost crabs are ectotherms [1]; habitat type, to assess whether activity varied across habitats with different vegetative communities and disturbance on the barrier island landscape [1]; level of shell cover, to account for variation in surface armoring which may affect the ability of ghost crabs to burrow; and point type (i.e., random sampling location versus a piping plover or American oystercatcher nest), to assess whether activity was higher at nests relative to random sampling locations [4]. We investigated the importance of all variables on ghost crab abundance as fixed additive effects. Additionally, we investigated the importance of two fixed interactive effects: first, the interaction between temperature and date, to account for how the effect of temperature on burrow abundance may depend on date; and second, the interaction between point type and shell cover, as shorebirds may select nesting habitat that is disadvantageous for ghost crab burrowing activity [1, 37].

We completed all analyses in the statistical computing environment R ver. 4.2.1 [38]. We fit models using generalized linear mixed-effects models (GLMMs) using the package *glmmTMB* [39]. As our response variable (i.e., ghost crab burrow abundance) contained many zeros and was overdispersed, we fit zero-inflated negative binomial models [40]. The GLMMs contained both a zero-inflated component, which predicted the probability of observing extra zeros in the burrow count data, and a conditional component, which predicted ghost crab burrow counts conditional on the zero-inflated model [40] (e.g., [41]). In all of our candidate GLMMs, the zero-inflated component contained only sampling date as a fixed effect, as the absence of ghost crab burrow (i.e., zero count) was most common early in the season before the start of the ghost crab mating and recruitment period in June [1]. The conditional component contained a random effect for sampling point, to account for repeated measurements at each

point, as well as one or more variables as fixed effects (S1 Table). Our global model contained the zero-inflated component for sampling date, and the conditional component included the random effect for sampling point, all five predictor variables as additive effects, and the two interactions. We checked all standard model assumptions using the packages *DHARMa* [42] and *spdep* [43] and verified that there was no concerning spatial autocorrelation.

We used an information-theoretic approach to model selection, in which we compared 26 candidate models and selected a top model to use for inference [44–46]. Our model set included the global model, a null model, and 25 additional candidate models containing different combinations of the fixed effect variables included in the conditional portion of the global model, based on *a priori* hypotheses about the effects of each predictor variable on burrow abundance (S1 Table). We used the package *AICcmodavg* [47] to rank the 26 candidate models using second-order Akaike's information criterion corrected for sample size ($AIC_c$) [48]. We used $\Delta AIC_c$ and Akaike weights ($\omega_i$) to assess model support [46, 49]. We considered lower-ranked candidate models to be equally plausible as the top-ranked model when $\Delta AIC_c \leq 2$. [49].

Using the top-ranked model, we interpreted variables with a positive beta estimate as having a positive effect on ghost crab burrow abundance and variables with a negative beta estimate as having a negative effect on ghost crab burrow abundance [50]. Additionally, we interpreted the magnitude of beta estimates as the strength of the effect on ghost crab burrow abundance [50]. We identified informative variables from the top ranked model as those parameters for which the 95% confidence interval did not include zero [51, 52].

### Ethics statement

All research was conducted in compliance with the laws of the United States of America, followed the protocols of the Chincoteague National Wildlife Refuge Research and Monitoring Special Use Permit (# 2022–004) and The Nature Conservancy in Virginia Research Permit (approval issues in 2022), and were approved by Virginia Tech's Institutional Animal Care and Use Committee (protocol # 19–248 FWC). Additionally, as the piping plover is a protected species, we followed all protocols of the Virginia Department of Wildlife Resources Threatened/Endangered Species Permit (ID # 2558265). The island sampled is owned and managed by The Nature Conservancy in Virginia and the United States Fish and Wildlife Service. The Nature Conservancy in Virginia, the United States Fish and Wildlife Service, and the Virginia Department of Wildlife Resources are responsible for all scientific and ecological monitoring of the island. Public access to the island is seasonally restricted by the managing organization and agency, so that the public cannot access land located above the intertidal zone or outside of designated island crossover pathways. All applicable ethical guidelines for the use of animals in research have been followed, including those presented in the Ornithological Council's *Guidelines to the Use of Wild Birds in Research* [53].

### Results

We conducted surveys at 38 random sampling locations and 63 nests (piping plover *n* = 19, American oystercatcher *n* = 44) on Metompkin Island from 24 April to 17 July 2022. At random sampling locations, burrow counts ranged from 0–55 burrows/point (mean = 7.85 ± 0.34 SE) and at nests, burrow counts ranged from 0–43 burrows/point (mean = 6.41 ± 0.39 SE). On average, burrow counts were greater at American oystercatcher nests than piping plover nests (mean ± SE = 6.56 ± 0.41 burrows/American oystercatcher nest versus 5.45 ± 1.18 burrows/piping plover nest).

Our global model, which included an interaction between date and temperature, an interaction between point type and shell cover, and all five additive effects, was our top model for

explaining ghost crab burrow presence and abundance ($\omega_i = 0.71$; Table 1). One other model, which contained date, temperature, point type, shell cover, and habitat type as additive effects only, was nearly competitive ($\Delta AIC_c = 2.33$; Table 1).

The probability of ghost crab burrow absence at a sampling point (i.e., a zero count) decreased as the shorebird breeding season progressed ($\beta_{zi\ date} \pm SE = -1.87 \pm 0.34$). The interaction between date and temperature was not an informative predictor of ghost crab burrow counts ($\beta = -0.06$, 95% CI = -0.13–0.01; Fig 3). Ghost crab burrow counts were lower at piping plover nests with both sparse and heavy shell cover, relative to random sampling locations with no shell cover ($\beta \pm SE = -1.59 \pm 0.58$ for piping plover nests with sparse shell cover and $\beta$

**Table 1. Candidate models examining the factors influencing Atlantic ghost crab (*Ocypode quadrata*) burrow abundance on Metompkin Island, Virginia.** The fixed effects from the conditional model component in each candidate model are listed. The use of '+' indicates additive terms in a multivariable model, and the use of '×' indicates an interaction. All models (including the null model) include a random effect for sampling point and a zero-inflated component for sampling date. Models are ranked by ascending $\Delta AIC_c$.

| Conditional model component[a] | $k$[b] | LL[c] | $\Delta AIC_c$[d] | $\omega_i$[e] |
|---|---|---|---|---|
| Habitat + point type + shell cover + date + temperature + point type × shell cover + date × temperature | 22 | -2,304.51 | 0.00 | 0.71 |
| Habitat + point type + shell cover + date + temperature | 17 | -2,310.92 | 2.33 | 0.22 |
| Habitat + shell cover + date + temperature + date × temperature | 16 | -2,314.01 | 6.41 | 0.03 |
| Habitat + shell cover + temperature + date | 15 | -2,315.16 | 6.65 | 0.03 |
| Habitat + point type + date + temperature + date × temperature | 16 | -2,314.88 | 8.15 | 0.01 |
| Habitat + date + temperature + date x temperature | 14 | -2,317.76 | 9.77 | 0.01 |
| Habitat + shell cover + temperature | 14 | -2,318.96 | 12.17 | < 0.01 |
| Point type + shell cover + date + temperature + point type × shell cover + date × temperature | 16 | -2,318.93 | 16.26 | < 0.01 |
| Shell cover + date + temperature + date × temperature | 10 | -2,328.47 | 22.94 | < 0.01 |
| Point type + date + temperature + date × temperature | 10 | -2,331.64 | 29.29 | < 0.01 |
| Date + temperature + date × temperature | 8 | -2,334.84 | 31.58 | < 0.01 |
| Temperature | 6 | -2,339.97 | 37.78 | < 0.01 |
| Habitat + date + point type + shell cover + point type × shell cover | 20 | -2,331.19 | 49.14 | < 0.01 |
| Habitat + point type + date | 14 | -2,340.55 | 55.35 | < 0.01 |
| Habitat + date | 12 | -2,344.56 | 59.24 | < 0.01 |
| Point type + shell cover + date + point type × shell cover | 14 | -2,348.30 | 70.85 | < 0.01 |
| Habitat + point type + shell cover + point type × shell cover | 19 | -2,343.52 | 71.71 | < 0.01 |
| Shell cover + date | 8 | -2,356.83 | 75.56 | < 0.01 |
| Point type + date | 8 | -2,360.17 | 82.25 | < 0.01 |
| Habitat + point type | 13 | -2,355.37 | 82.91 | < 0.01 |
| Date | 6 | -2,362.81 | 83.46 | < 0.01 |
| Habitat | 11 | -2,362.34 | 92.74 | < 0.01 |
| Point type + shell cover + point type × shell cover | 13 | -2,361.74 | 95.66 | < 0.01 |
| Shell cover | 7 | -2,372.27 | 104.41 | < 0.01 |
| Point type | 7 | -2,374.98 | 109.83 | < 0.01 |
| Null (Intercept-only) | 5 | -2,378.00 | 111.80 | < 0.01 |

[a] Fixed effect variables include date, habitat (i.e., intertidal zone, berm, beach, overwash, dunes, backdune flat, and backbarrier), level of shell cover within a 2 m radius of the survey point (i.e., none, sparse, or heavy), point type (i.e., random sampling location versus piping plover or American oystercatcher nest), and mean daily air temperature.

[b] $k$ = the number of parameters in the model

[c] LL = log likelihood

[d] $\Delta AIC_c$ = the difference between the model's Akaike's Information Criterion (corrected for small sample size) and that of the top-ranked model. $AIC_c$ for the top-ranked model was 4709.21

[e] $\omega_i$ = Akaike model weight

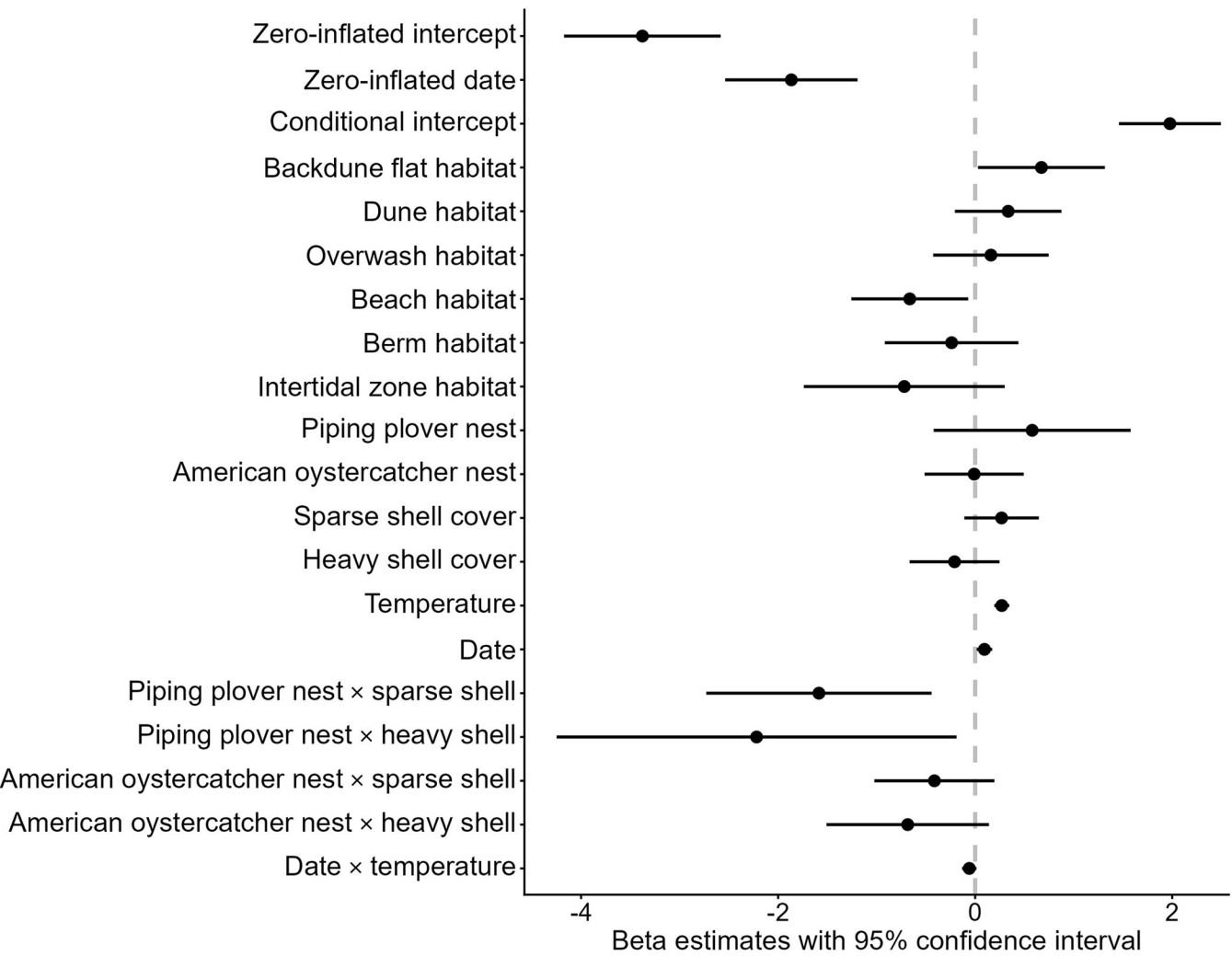

**Fig 3. Beta estimates with 95% confidence intervals for variables in the top-ranked model for predicting Atlantic ghost crab (*Ocypode quadrata*) burrow abundance on Metompkin Island, Virginia.** A positive effect indicates burrow abundance increases as the explanatory variable increases, whereas a negative effect indicates burrow abundance decreases as the explanatory variable increases. The model included three categorical fixed effects. 'No shell cover' was the reference level for the shell cover variable. 'Backbarrier habitat' was the reference level for the habitat variable. 'Random sampling location' was the reference level for American oystercatcher and piping plover nests.

± SE = -2.22 ± 1.04 for piping plover nests with heavy shell cover; Fig 3), though we did not find evidence of the same pattern at American oystercatcher nests (Fig 3).

Ghost crab burrow counts were greater in the backdune flat habitat, relative to the backbarrier habitat (β = 0.67 ± 0.33). In contrast, burrow counts were lower on the beach, relative to backbarrier habitat (β = -0.66 ± 0.30). No other habitat type was an informative predictor of ghost crab burrow abundance relative to the backbarrier habitat. Full results of the top-ranked model are in S2 Table.

## Discussion

Ghost crabs are known predators of shorebird eggs and chicks [3–7]. However, studies on their role in sandy beach ecosystems that include beach-nesting shorebirds are limited (e.g., [9,

54]). Understanding spatiotemporal patterns of ghost crab activity can be a useful tool for planning management actions to protect breeding shorebirds.

Atlantic ghost crabs are known to occupy the full profile of sandy beaches [1], and fine-scale habitat characteristics may affect where they are most abundant on the landscape. On Metompkin Island, ghost crab activity was greater in backdune flats, and lower on the beaches, relative to the island backbarrier, which we expected to have little to no evidence of ghost crab activity due to the presence of dense vegetation that obstructs burrow construction. This pattern may, in part, reflect the dynamic nature of barrier islands. Burrows in backdune flats are likely protected from high tides and storm surge by the dunes, and in substrate at least partially stabilized by vegetation. In contrast burrows on the beach are more ephemeral and susceptible to collapse following overwash from waves.

We found ghost crab activity was lower at piping plover nest sites with sparse and heavy shell cover, relative to random locations with no shell cover. However, we did not find evidence that this same pattern existed for American oystercatchers. This could be explained by habitat requirements and behavioral interactions between ghost crabs and shorebirds.

Both piping plovers and American oystercatchers select nest sites based on substrate characteristics that provide camouflage for their eggs, such as shell and wrack [32, 33, 37]. Though both species select similar habitats at the landscape level, their microhabitat preferences differ, with piping plovers selecting nest substrate with greater proportions of shell cover relative to American oystercatchers [37]. In contrast, ghost crabs are generally associated with soft sand that is less likely to restrict burrowing activity and movement [1, 55]. Thus, if piping plovers on Metompkin Island are mostly nesting on shell-armored substrate that obstructs crab burrow construction, we can expect lower burrow counts around piping plover nest sites.

Additionally, ghost crabs may further be deterred from constructing burrows near piping plover nests, as piping plovers will aggressively defend their nests from potential threats [32], which has been previously observed at Metompkin Island [28]. Other plover species, such as Wilson's plovers (*Charadrius wilsonia*), exhibit anti-predator behaviors in response to ghost crabs [56]. Similarly, piping plovers may use anti-predator strategies to defend their territory against ghost crabs, thus reducing the level of ghost crab activity immediately around their nest. In contrast, American oystercatchers often flush from their nests when a predator intrudes the nest site and use distraction behaviors to draw predator attention away from the nest [33]. Evidence also suggests American oystercatchers may not react at all when ghost crabs intrude the nest site [57].

Environmental factors also affect when ghost crabs are most likely to be active during the shorebird breeding season. We found that the probability of no ghost crab activity (i.e., no ghost crab burrows) at a sampling point declined as the shorebird breeding season progressed, and thus the probability of ghost crabs being active at a sampling point was greatest later in the breeding season. In the Mid-Atlantic, ghost crabs are most active during their mating and recruitment period in summer and fall (e.g., June through November) [1]. This period of expected peak activity for ghost crabs overlaps with the incubation and re-nesting period for American oystercatchers and piping plovers, from April to mid-June and mid-April to July, respectively [58–60]. Assuming the same nesting phenology that has been historically observed in Virginia and the Mid-Atlantic for these shorebird species, the greatest ghost crab activity should occur when there are fewer shorebird nests across the landscape.

A potential shift in the phenology of ghost crab activity or shorebird nesting in response to climatic trends may disrupt this pattern, potentially altering predator-prey interactions between shorebirds and ghost crabs [61]. While we did not find evidence of the interaction between date and temperature to be an important predictor of ghost crab activity in this study, prior research indicates ghost crabs in the United States Mid-Atlantic require temperatures

above 15˚C to be active [1]. Thus, warmer daily air temperatures occurring earlier in the season due to global climate change may result in increased ghost crab activity during the shorebird nesting season. Additionally, shorebirds demonstrate variable breeding phenology in response to changing climatic conditions [62]. Even if shorebird species maintain similar timing for nest initiation, altered patterns of storminess and increased flooding in the Mid-Atlantic (e.g., lengthened winter storm season and recurrent flooding as a result of the combined effects of sea-level rise and storm surge; [63, 64]) may lead to changes in shorebird re-nesting patterns, resulting in a longer nesting period and delay in chick hatching. If ghost crab activity peaks earlier in the season or shorebird nesting patterns shift to later in the season, ghost crabs are more likely to overlap with shorebird nests and chicks in the environment, increasing the potential predation threat to shorebird reproductive success.

Despite documentation of ghost crab predation of shorebird nests on Metompkin Island [28], we did not find evidence to suggest that ghost crab activity was greater at shorebird nest sites relative to random sampling locations. However, ghost crab activity was still detected at all of the monitored nests at some point during the breeding season, suggesting that the potential predation threat remains. Additionally, shorebird chicks are precocial [32, 33], meaning that they will quickly leave the nest site after hatching and the tending adults may lead them to areas with different levels of ghost crab activity. If phenological shifts result in shorebird nesting and chick-rearing increasingly overlapping with peak ghost crab activity on the landscape, the relative threat that ghost crabs pose to nesting shorebirds may increase. Land managers should consider implementing ghost crab population monitoring protocols to document patterns in population abundance and distribution and, where warranted, begin developing management strategies on beaches where ghost crabs are believed to be a limiting factor to reproductive success.

Further research is needed to understand how habitat characteristics and environmental factors influence spatiotemporal ghost crab activity and overlap with shorebird breeding phenology across sites. Understanding microhabitat characteristics selected by both ghost crabs and nesting shorebirds will help to identify practical and effective strategies for managing the threat of ghost crabs to shorebird reproductive success. For example, shell placement could promote successful shorebird nesting by limiting ghost crab burrow construction and reducing ghost crab activity at nest sites. Additionally, further investigation into the potential for a finer-scale association between temperature, date, and crab burrow abundance and for a phenological shift in either ghost crab activity or shorebird reproduction may help to assess if the ghost crab predation risk to shorebird eggs and chicks is evolving in response to climate change.

## Supporting information

**S1 Table. Conditional components of the 26 candidate models to predict the abundance of ghost crab burrows within a 2m radius of sampling points on Metompkin Island, Virginia.** Fixed effects varied between models, based on *a priori* hypotheses about factors thought to affect ghost crab activity. The use of '+' indicates additive terms in a multivariable model, and the use of '×' indicates an interaction between two terms. All models include a random effect for sampling point. Additionally, all models include a zero-inflated component for sampling date. The response variable is ghost crab burrow abundance.
(DOCX)

**S2 Table. Full results of the top-ranked model from the model selection procedure.** The beta estimate, standard error, and 95% confidence interval for each parameter in the top-ranked model are listed.
(DOCX)

## Acknowledgments

We thank staff at the University of Virginia Coastal Research Station, as well as S. Ritter and D. Fraser from the Virginia Tech Department of Fish and Wildlife Conservation, for their logistical and intellectual support over the course of the study. We also thank biologists and technicians from The Nature Conservancy Virginia Coast Reserve and Virginia Department of Wildlife Resources who may have assisted with field data collection. Virginia Tech acknowledges that we work on the Tutelo/Monacan People's homeland, and we recognize their continued relationships with their lands and waterways. We further acknowledge that the Morrill Land-Grant College Act (1862) enabled the Commonwealth of Virginia to finance and found Virginia Tech through the forced removal of Native Nations from their lands in California and other areas in the West. We also acknowledge that during this research, we lived and worked on Pocomoke, Occohannock, and Accomack Peoples' homeland and we recognize their continued connection to the land and water of the Virginia coast. Further, we acknowledge that our work done at Virginia Tech in Blacksburg is on land that was previously the site of the Smithfield and Solitude Plantations, owned by members of the Preston family. Between the 1770s and the 1860s, the Prestons and other local white families that owned parcels of what became Virginia Tech also owned hundreds of enslaved people. Enslaved Black people generated resources that financed Virginia Tech's predecessor institution, the Preston and Olin Institute, and they also worked on the construction of its building.

## Author Contributions

**Conceptualization:** Mikayla N. Call, Sarah M. Karpanty, Kristy C. Lapenta, Alexandra L. Wilke, Ruth Boettcher.

**Data curation:** Mikayla N. Call, Rasheed S. Pongnon, Kristy C. Lapenta.

**Formal analysis:** Mikayla N. Call, Rasheed S. Pongnon, Christy N. Wails, Sarah M. Karpanty.

**Funding acquisition:** Mikayla N. Call, Rasheed S. Pongnon, Sarah M. Karpanty, James D. Fraser.

**Investigation:** Mikayla N. Call, Rasheed S. Pongnon, Kristy C. Lapenta, Alexandra L. Wilke, Ruth Boettcher.

**Methodology:** Mikayla N. Call, Sarah M. Karpanty, Kristy C. Lapenta, Alexandra L. Wilke, Ruth Boettcher.

**Project administration:** Mikayla N. Call, Christy N. Wails, Sarah M. Karpanty, Kristy C. Lapenta.

**Software:** Mikayla N. Call, Rasheed S. Pongnon, Christy N. Wails.

**Supervision:** Mikayla N. Call, Sarah M. Karpanty, Kristy C. Lapenta.

**Validation:** Mikayla N. Call, Christy N. Wails, Sarah M. Karpanty.

**Visualization:** Mikayla N. Call, Kristy C. Lapenta, Camille R. Alvino.

**Writing – original draft:** Mikayla N. Call, Rasheed S. Pongnon, Sarah M. Karpanty.

**Writing – review & editing:** Mikayla N. Call, Rasheed S. Pongnon, Christy N. Wails, Sarah M. Karpanty, Kristy C. Lapenta, Alexandra L. Wilke, Ruth Boettcher, Camille R. Alvino, James D. Fraser.

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
