## [Decision Letter · Decision Letter 0]

29 May 2024

PONE-D-24-09865Biotic and abiotic factors affecting Atlantic ghost crab (* Ocypode quadrata* ) spatiotemporal activity at an important shorebird nesting site in VirginiaPLOS ONE

Dear Dr. Call,

Thank you for submitting your manuscript to PLOS ONE. After careful consideration, we feel that it has merit but does not fully meet PLOS ONE’s publication criteria as it currently stands. Therefore, we invite you to submit a revised version of the manuscript that addresses the points raised during the review process.

The major issues with your submission are in your analytical approach and interpretation of results. Publications in PLOS ONE are expected to present analyses that are conducted to a high technical standard and with enough detail to be reproducible. Two reviewers have provided detailed comments below to help you with your analyses and revisions to the manuscript.

We look forward to receiving your revised manuscript.

Kind regards,

Stephanie S. Romanach, Ph.D.

Academic Editor

PLOS ONE

Reviewers' comments:

Reviewer's Responses to Questions

**Comments to the Author**

1. Is the manuscript technically sound, and do the data support the conclusions?

Reviewer #1: Partly

Reviewer #2: Yes

2. Has the statistical analysis been performed appropriately and rigorously? 

Reviewer #1: No

Reviewer #2: No

3. Have the authors made all data underlying the findings in their manuscript fully available?

Reviewer #1: Yes

Reviewer #2: No

4. Is the manuscript presented in an intelligible fashion and written in standard English?

Reviewer #1: Yes

Reviewer #2: Yes

5. Review Comments to the Author

Reviewer #1: "I would appreciate a revision of the map in Fig. 1, which could display less of the USA but should provide much higher detail of Metompkin Island. Indeed, readers of the paper may be interested in better visualizing the study area. Therefore, I suggest zooming in much more on that island. It's not even necessary to show the entire island, but the position of the surveyed area should be identifiable on the map. If this proves to be practically impossible, at least the coordinates of the limits of the 3 km-long stretch of the island used for the survey should be added to ensure the research can be fully replicated in the future by other researchers.

In the Abstract, it is stated, "We observed burrows at all nest sites in our study area but found fewer burrows at nest sites than random sites." This was an unexpected result for the authors. Consequently, they suggested that "Ghost crabs may avoid shorebird nest sites due to aggressive defensive behaviors from incubating adults or differences in microhabitat characteristics selected by shorebirds versus ghost crabs." This point is intriguing for readers because the study was not solely motivated by interest in ghost crab behavior and ecology but primarily due to the threat posed by the crabs to the reproductive success of endangered birds. Therefore, it's expected that the available data would be utilized, if possible, to test which of the two suggested causes was involved. While the behavior of the nesting birds wasn't studied, as far as I'm aware, the microhabitat was examined. Microhabitat is significant because other studies have demonstrated that piping plovers prefer to nest in areas with above-average shell cover, while crabs prefer to dig in areas with a low amount of shells, where digging is easier for them. However, the interaction between the presence of shorebird nests and the amount of shell cover wasn't tested.

I speculate that this issue could be investigated by incorporating, in the glmm model, a dependent variable: the interaction term between substrate type and the presence of bird nests. If there was a reason not to test this interaction, authors should explain why. As a reader, I'd also like to understand the extent to which the increase in the number of burrows depends on the likely interaction of date and temperature. Therefore, I suggest adding this interaction term to the glmm model as well. I understand that the authors obviously didn't include all possible interaction terms between predictor variables and that they chose to test only a set of 17 models based on a priori hypotheses about the effect of each predictor variable, but the suggested two interactions seem ecologically sound.

Additionally, as a reader, I'm curious to know whether the diameter of the burrows was measured, in addition to the number of burrows. I presume that the size of crabs would be significant in their interaction with birds. Large ghost crabs pose more danger to eggs and chicks, while tiny crabs with a carapace length of 1 cm should not be dangerous or could even be considered as food for birds. Therefore, was the size of the burrows measured? If not, I expect the authors to add a brief note in the Methods section explaining why it wasn't done despite its potential significance.

Not much information is provided on the glmm used for the analysis. For example, which response distribution and error distribution were used? Please add all relevant information that can be useful for the reader.

As I'm not an expert in statistics, I would appreciate it if the authors could provide a key in the Methods section on how to evaluate when an effect size is considered significant. For example, how significant are the apparently very low beta values of 0.05 and 0.08 for temperature and date (line 189)? Again, I'm not an expert in statistics, but I wonder why the beta effect sizes of dune and back dune flat habitat reported in the Results (0.05+-0.01 and 0.05+-0.01) (line 191) do not correspond to what can be seen in Fig. 2. Please correct or explain. Is the SE of the effect size of piping plover nest really 0.01 as stated in the results (line 214)? It appears to be different in Fig. 2. Please verify.

Regarding the Acknowledgments, I find it peculiar that "In addition to the authors listed for this study, we thank staff at the University of Virginia Coastal Research Station." Does this mean that the authors are thanking themselves? Later, D. Fraser is thanked. Is he the same D. Fraser who is one of the co-authors?"

Reviewer #2: Overall impression:

This study on Atlantic ghost crabs and their potential impact on beach-nesting shorebirds is well-written and addresses an important, yet under-appreciated topic of conservation interest. I have some concerns regarding the analytical methods, however, which could be addressed with the addition of significant clarification and justification in the manuscript. Specifically, while the abstract outlines the methodology and key findings regarding ghost crab activity and habitat selection, a more thorough account of hypothesis testing and statistical significance would enhance the manuscript's rigor. Overall, the study contributes valuable insights into ghost crab ecology and its implications for shorebird conservation, but further refinement of the analysis and interpretation is needed to strengthen the manuscript.

Major comments:

1) The authors should provide further clarification on the methods used for counting burrows between the random point locations and the nest sites. If random point surveys were indeed a point, then a 2m radius around the outside of a nest would yield a greater surface area (based on basic geometric principles) for examination compared to a 2m radius around a point, correct? If nest size varies greatly, this would also cause variability in surface area for surveys. Perhaps the diameter of the nests are small enough that the difference in area would be negligible, but the authors should address this possibility and provide further clarification on the characteristics of shorebird nest size.

2) The abstract and introduction state that the authors tested whether ghost crab activity was greater at shorebird nest sites than random sites, however the results presented in the manuscript do not adequately answer this question and lack testing with any level of statistical significance. While the model procedure used for analysis seems appropriate, the text is lacking proper explanation of full analysis procedures. Additionally, it does not appear that the authors have conducted any significance tests (e.g. Wald tests or likelihood ratio tests) along with their models. Upon inspection of Figure 2, it appears that many of the model effects are not statistically significant (i.e. the confidence intervals include zero). If this is not the proper interpretation, the authors should take care to provide further details and explanation in the methods and results sections.

Minor comments:

Line 23— The first line of the abstract makes it sound like ghost crabs prey on hatched chicks and adult shorebirds, but it is clear in the other text that they are potential predators only of nests and eggs. Please clarify here to be in agreement with other sections of the text.

Line 52— causes used twice, please replace one instance

Line 60— reword to “are not often targeted.” Are the authors aware of any instances where ghost crabs are targeted, aside from the one report citation from FL?

Line 62—change to “evidence” from isolated studies

Line 70— are conditions “inhospitable” or simply outside the preferred or optimal temperature range? Consider word use here, perhaps “suboptimal” would be a better fit?

Line 72— what kind of “change” in population is being referred to here. Please be more explicit.

Line 80— change “spatiotemporal” to spatial and temporal

Methods

Line 107— Please, remove one instance of “species” to avoid repetition

Line 130— Do the shorebird nests vary in size/diameter? This would affect the total area of surveys when using a 2m radius around a nest. Similarly for the point surveys. Please see major comment #1 above.

Statistical analysis

Line 166— What type of mixed effects model? In this case, “global” is used improperly and the type of model should be explicitly stated. I assume you are referring to a generalized linear mixed-effects model (GLMM)? The distribution and function family used in the GLMM should also be clearly described. Please, define your global model with all covariates in a separate sentence, with each variable explicitly listed as a fixed or random effect along with relevant interaction terms.

From the current analysis description, it is unclear whether you tested different distributions (e.g. negative binomial, poisson, or ordinary least squares) in the process of model selection.

Line 172— what are the a priori hypotheses that were tested in the candidate models? This is unclear based on the predictions outlined in the introduction and should be explicitly stated. “Considering what is known about ghost crab ecology” is a vague rationale for the choice of candidate models and additional text is needed here to explain and justify model construction.

Results

Lines 182-183— is it not clear that the parentheticals contain the mean values, please include “mean” for each.

Lines 188-191 – Please edit the parenthethicals containing statistical results to explicitly include SE and p-values , where appropriate for fixed effects.

Line 189— when examining Fig. 2, it does not appear that these are significant terms in model (date and temperature). Can the authors please explain?

Line 190— if habitat was included as a fixed effect in the model, how are you reporting statistical results for levels within habitat? This is not clear in the analysis methods, and if post hoc tests with multiple comparisons were made then this should be explicit.

Discussion

Line 248— delete “However”

Line 271— change “may need to” to “should”

Line 274— authors should be clear what they mean by “future research” directions. For example, it seems like it would be helpful to have data that could potentially help capture phenological shifts in predation risk. The authors should also consider including a final conclusion sentence (or two) that highlights the strength of their contribution and how this study can provide a basis for these future research in this area.

Line 275—change to “spatiotemporal ghost crab activity” as the crab is the noun in this sentence.

Figures and Tables

Table 1— It is not necessary to identify all levels for each variable of the model in the caption, but the variables should be clearly listed as fixed factors. If not explicitly including the random effect on each line, the caption should read “All models included a random effect of sampling point. Additionally, the code is not available so it is not possible to check whether the random effect was coded properly.

The manuscript has minimal figures. While I think the present figures/tables are effective, I suggest that the authors consider including an image figure to show nest sites and random point sites, highlighting the burrows that were counted. This could be included in the main text given the current low figure count, or included as an appendix.

6. PLOS authors have the option to publish the peer review history of their article (what does this mean?). If published, this will include your full peer review and any attached files.

Reviewer #1: No

Reviewer #2: No

---

## [Author Response · Author response to Decision Letter 0]

9 Jul 2024

Please see our 'Response to Reviewers' document for responses to specific reviewer and editor comments. Those responses have been included below (with some modification to accommodate formatting). 

Editorial Comments and Responses

We thank the two reviewers and Academic Editor for their thoughtful comments and constructive criticism which have improved this manuscript. We provide our responses to the journal requirements and all reviewer comments in italics below.

Thank you for providing these resources. We have reviewed the submission guidelines and have ensured that the manuscript meets PLOS ONE’s style requirements outlined in the provided links. 

Since submission of the manuscript to PLOS ONE, we have made the data publicly available via the Environmental Data Initiative (EDI) Data Portal. The data was published on 15 March 2024 and is available for download here: https://doi.org/10.6073/pasta/1f1117a917de507802f1acb4d5cf42ef

We have entered the full Data Availability Statement that includes the name of the database and location of the data in the PLOS ONE submission form. Additionally, we have cited the data in the methods section on line 181.

Thank you for the opportunity to improve the clarity on the ethics of our study. As requested, we have included our full ethics statement in the ‘Methods’ section of our manuscript file on lines 230–245. To summarize:

This project was an observational field study that included observations of some vertebrate species (i.e., shorebirds), one of which is a federally- and state-protected species (i.e., the piping plover Charadrius melodus). We have revised our ethics statement to include the required information for both animal research and observational and field studies, as specified in the submission guidelines for PLOS ONE. Our ethics statement includes: 

• The full names and numbers, where applicable, as well as the full name of the issuing agency/organization, for all permits under which we conducted the research. (This was included in the ethics statement from the original submission and is unchanged). 

• The full name of the IACUC (i.e., Institutional Animal Care and Use Committee) office that approved the research, and the associated protocol number. (This was included in the ethics statement from the original submission and is unchanged). 

• A statement on how one species in our study, the piping plover, is a protected species. (This was added in revisions on lines 235–237). 

• A statement on landownership for the study site, including that the island is jointly owned and managed by a private non-governmental organization (i.e., The Nature Conservancy) and a federal land management agency (i.e., U.S. Fish and Wildlife Service). (This was included in the ethics statement from the original submission and is unchanged). Additionally, we include a statement that public access to the site is seasonally restricted by the managing organization/agency so that the public cannot access land above the intertidal zone or outside designated pathways. (This was added in revisions on lines 241–243). 

• A statement about how our research followed all applicable ethical guidelines for research involving wild avian species, as presented in the Ornithological Council’s Guidelines to the Use of Wild Birds in Research. (This was included in the ethics statement from the original submission; however, we included a citation for the Guidelines in revisions on line 245 and provide the full citation in reference 53 on lines 565–567).

We do not include an IRB (i.e., Institutional Review Board) protocol number or statement on consent (written or verbal), as this project did not include human subject research and thus did not involve human study participants.

a. You may seek permission from the original copyright holder of Figure 1 to publish the content specifically under the CC BY 4.0 license. We recommend that you contact the original copyright holder with the Content Permission Form (http://journals.plos.org/plosone/s/file?id=7c09/content-permission-form.pdf) and the following text: “I request permission for the open-access journal PLOS ONE to publish XXX under the Creative Commons Attribution License (CCAL) CC BY 4.0 (http://creativecommons.org/licenses/by/4.0/). Please be aware that this license allows unrestricted use and distribution, even commercially, by third parties. Please reply and provide explicit written permission to publish XXX under a CC BY license and complete the attached form.” Please upload the completed Content Permission Form or other proof of granted permissions as an "Other" file with your submission. In the figure caption of the copyrighted figure, please include the following text: “Reprinted from [ref] under a CC BY license, with permission from [name of publisher], original copyright [original copyright year].”

b. If you are unable to obtain permission from the original copyright holder to publish these figures under the CC BY 4.0 license or if the copyright holder’s requirements are incompatible with the CC BY 4.0 license, please either i) remove the figure or ii) supply a replacement figure that complies with the CC BY 4.0 license. Please check copyright information on all replacement figures and update the figure caption with source information. If applicable, please specify in the figure caption text when a figure is similar but not identical to the original image and is therefore for illustrative purposes only. The following resources for replacing copyrighted map figures may be helpful: USGS National Map Viewer (public domain): http://viewer.nationalmap.gov/viewer/, The Gateway to Astronaut Photography of Earth (public domain): http://eol.jsc.nasa.gov/sseop/clickmap/, Maps at the CIA (public domain): https://www.cia.gov/library/publications/the-world-factbook/index.html and https://www.cia.gov/library/publications/cia-maps-publications/index.html, NASA Earth Observatory (public domain): http://earthobservatory.nasa.gov/, Landsat: http://landsat.visibleearth.nasa.gov/, USGS EROS (Earth Resources Observatory and Science (EROS) Center) (public domain): http://eros.usgs.gov/#, Natural Earth (public domain): http://www.naturalearthdata.com/

Thank you for the additional information regarding PLOS ONE’s policy related to publishing copyrighted maps or satellite imagery. We checked the copyright status of all data included in Figure 1. Figure 1 includes the following data: 

• A basemap of the United States obtained from the U.S. Census Bureau’s Topologically Integrated Geographic Encoding and Referencing (TIGER) database. These data have no access constraints, are not copyrighted, and are free to use in publication as long as acknowledgement is given to the U.S. Census Bureau. 

The metadata for the basemap (https://catalog.data.gov/dataset/tiger-line-shapefile-current-nation-u-s-state-and-equivalent-entities) explicitly states: “Use Constraints: The TIGER/Line Shapefile products are not copyrighted however TIGER/Line and Census TIGER are registered trademarks of the U.S. Census Bureau. These products are free to use in a product or publication, however acknowledgement must be given to the U.S. Census Bureau as the source. The boundary information in the TIGER/Line Shapefiles are for statistical data collection and tabulation purposes only; their depiction and designation for statistical purposes does not constitute a determination of jurisdictional authority or rights of ownership or entitlement and they are not legal land descriptions. Coordinates in the TIGER/Line shapefiles have six implied decimal places, but the positional accuracy of these coordinates is not as great as the six decimal places suggest.” 

As per the constraints, we provide acknowledgement to the U.S. Census Bureau in the caption for Figure 1 on line 102.

• A basemap of Virginia obtained from the Virginia Geographic Information Network (VGIN) Virginia GIS Clearinghouse via the Virginia Administrative Boundaries download (https://vgin.vdem.virginia.gov/datasets/777890ecdb634d18a02eec604db522c6/about). We have confirmed with a representative from VGIN that all data distributed through the Virginia GIS Clearinghouse can be used and redistributed without restriction or license. We have upload proof of correspondence as an ‘Other’ file with our submission, as requested. 

5. Review Comments to the Author

Reviewer #1: 

"I would appreciate a revision of the map in Fig. 1, which could display less of the USA but should provide much higher detail of Metompkin Island. Indeed, readers of the paper may be interested in better visualizing the study area. Therefore, I suggest zooming in much more on that island. It's not even necessary to show the entire island, but the position of the surveyed area should be identifiable on the map. If this proves to be practically impossible, at least the coordinates of the limits of the 3 km-long stretch of the island used for the survey should be added to ensure the research can be fully replicated in the future by other researchers.

Thank you for this comment, as we think your suggestions greatly improve this figure. We’ve revised Figure 1 according to the suggested revisions provided here. We have revised the inset map to show more detail of Metompkin Island and less detail of the United States. Additionally, we have indicated the location of the study area on the inset map.

Additionally, as suggested, we’ve provided the coordinates for the extent of the study area in-text in the study area description within the manuscript on lines 121–122. We also chose to provide coordinates for the centroid of the study area. 

In the Abstract, it is stated, "We observed burrows at all nest sites in our study area but found fewer burrows at nest sites than random sites." This was an unexpected result for the authors. Consequently, they suggested that "Ghost crabs may avoid shorebird nest sites due to aggressive defensive behaviors from incubating adults or differences in microhabitat characteristics selected by shorebirds versus ghost crabs." This point is intriguing for readers because the study was not solely motivated by interest in ghost crab behavior and ecology but primarily due to the threat posed by the crabs to the reproductive success of endangered birds. Therefore, it's expected that the available data would be utilized, if possible, to test which of the two suggested causes was involved. While the behavior of the nesting birds wasn't studied, as far as I'm aware, the microhabitat was examined. Microhabitat is significant because other studies have demonstrated that piping plovers prefer to nest in areas with above-average shell cover, while crabs prefer to dig in areas with a low amount of shells, where digging is easier for them. However, the interaction between the presence of shorebird nests and the amount of shell cover wasn't tested. I speculate that this issue could be investigated by incorporating, in the glmm model, a dependent variable: the interaction term between substrate type and the presence of bird nests. If there was a reason not to test this interaction, authors should explain why. As a reader, I'd also like to understand the extent to which the increase in the number of burrows depends on the likely interaction of date and temperature. Therefore, I suggest adding this interaction term to the glmm model as well. I understand that the authors obviously didn't include all possible interaction terms between predictor variables and that they chose to test only a set of 17 models based on a priori hypotheses about the effect of each predictor variable, but the suggested two interactions seem ecologically sound.

Thank you! We appreciate the time and thought that the reviewer put into formulating this suggestion, because we believe that including the two suggested interaction terms has the potential to improve the analysis and test our assumption of the mechanism behind our findings. 

The reviewer is correct that we did not quantify aggressive behavior of the nesting birds towards ghost crabs, as this study was not set up to allow for an observational study of nesting bird behavior. However, as the reviewer mentioned, we have the data to further investigate our hypothesis on differences between microhabitat characteristics used by ghost crabs vs nesting shorebirds. 

We revised our analysis based on the suggestions provided. First, we included an interaction term for point type and shell cover and an interaction term for date and temperature in our global model. We then revised our model set to now include 26 models, to account for adding two interaction terms to the global model. We have included a list of those models in S1 Table. We used the same model selection procedures as described in the prior manuscript draft to rank the models and select a top model. The global model (this time, including the interaction terms) came out as the top-ranked model. We considered one additional model (i.e., the model with all variables as additive effects only, and no interactive effects) to be competitive, but because the interaction effects of multiple variables are only in one of the top models, we did not choose to model average and only report parameter estimates from the top-ranked model. 

We’ve also revised the ‘Res

---

## [Editor Report · Decision Letter 1]

12 Jul 2024

Biotic and abiotic factors affecting Atlantic ghost crab (* Ocypode quadrata* ) spatiotemporal activity at an important shorebird nesting site in Virginia

PONE-D-24-09865R1

Dear Dr. Call,

We’re pleased to inform you that your manuscript has been judged scientifically suitable for publication and will be formally accepted for publication once it meets all outstanding technical requirements.

Kind regards,

Stephanie S. Romanach, Ph.D.

Academic Editor

PLOS ONE

---

## [Editor Report · Acceptance letter]

2 Aug 2024

PONE-D-24-09865R1 

PLOS ONE

Dear Dr. Call, 

I'm pleased to inform you that your manuscript has been deemed suitable for publication in PLOS ONE. Congratulations! Your manuscript is now being handed over to our production team.

Kind regards, 

on behalf of

Dr. Stephanie S. Romanach 

Academic Editor

PLOS ONE